# Robust Reinforcement Learning from Corrupted Human Feedback[*]

Alexander Bukharin[1]   Ilgee Hong[1]   Haoming Jiang[2]   Zichong Li[1]   Qingru Zhang[1]

Zixuan Zhang[1]                          Tuo Zhao[1]

## Abstract

Reinforcement learning from human feedback (RLHF) provides a principled framework for aligning AI systems with human preference data. For various reasons, e.g., personal bias, context ambiguity, lack of training, etc, human annotators may give incorrect or inconsistent preference labels. To tackle this challenge, we propose a robust RLHF approach – $R^3M$, which models the potentially corrupted preference label as sparse outliers. Accordingly, we formulate the robust reward learning as an $\ell_1$-regularized maximum likelihood estimation problem. Computationally, we develop an efficient alternating optimization algorithm, which only incurs negligible computational overhead compared with the standard RLHF approach. Theoretically, we prove that under proper regularity conditions, $R^3M$ can consistently learn the underlying reward and identify outliers, provided that the number of outlier labels scales sublinearly with the preference sample size. Furthermore, we remark that $R^3M$ is versatile and can be extended to various preference optimization methods, including direct preference optimization (DPO). Our experiments on robotic control and natural language generation with large language models (LLMs) show that $R^3M$ improves robustness of the reward against several types of perturbations to the preference data.

## 1   Introduction

As artificial intelligence (AI) systems continue to advance and become increasingly sophisticated, ensuring their alignment with human values and preferences has emerged as a paramount concern, particularly for recent large language models [37, 28]. One promising approach to achieving this alignment is Reinforcement Learning from Human Feedback (RLHF), which involves training AI systems through a process of reward modeling based on human-provided feedback and preferences [13, 3, 50].

A significant challenge in RLHF, however, arises from the inherent uncertainty present in the preference data provided by human evaluators [17, 4]. Since RLHF often targets highly complex scenarios where defining precise preference standards is difficult, if not impossible, annotators may provide undesirable or inconsistent preference labels, especially when they lack sufficient experience or training. In the case of a robotics system designed to assist with household tasks, an untrained annotator might label actions that complete the task efficiently but in a manner that could potentially cause property damage or compromise safety as preferable, overlooking the importance of safe and responsible operation.

An even more concerning scenario is that some human evaluators may maliciously assign incorrect preference labels [44, 32]. Personal prejudices, agendas, or lack of understanding about the true goals of the system could lead some annotators to intentionally mislabel examples, which could undermine the entire RLHF process and cause the model to learn undesirable or misaligned behaviors, posing a

---

[*]The authors are listed in alphabetical order: [1]Georgia Tech, [2]Amazon.

38th Conference on Neural Information Processing Systems (NeurIPS 2024).

significant risk to the robustness and reliability of the AI system. Despite its critical importance for AI system alignment, this issue has received limited attention in the existing literature. For example, when training an AI system for automated content moderation on social media platforms, malicious annotators could mislabel examples of hate speech, misinformation, or harmful content as desirable, leading the model to learn to allow the proliferation of toxic and dangerous online behaviors. Despite its critical importance for AI system alignment, the robustness of RLHF has only received limited attention in the existing literature [9].

To address this challenge, we propose a robust Reinforcement Learning from Human Feedback (RLHF) approach – $R^3M$ (**R**obust **R**eward **M**odeling for **R**LHF) to handle partially corrupted preference labels. Specifically, we assume that a subset of incorrect (corrupted) labels exists as outliers in the preference data used for training the reward model.[2] To model the label corruption, we introduce an instance-specific perturbation factor to the Bradley-Terry (BT) model for human preference [6]. We then learn the reward model and perturbation factors simultaneously by maximizing an $\ell_1$-regularized likelihood of the preference data. Theoretically, we prove that under proper regularity conditions, our approach can consistently learn the underlying ground truth reward and identify potential outliers, provided that the number of incorrect labels scales sublinearly with the preference sample size. Computationally, we show that the additional computational overhead of involving the perturbation factor in training is negligible: The log-likelihood is strictly convex and univariate with respect to each perturbation factor, and we can obtain its closed-form update at each iteration.

To demonstrate the effectiveness of our proposed method, we apply $R^3M$ to robotic control [39]. Specifically, we consider different types of corruptions to the preference data, including irrational flipping, stochastic flipping, and myopic flipping. We train robust reward models with $R^3M$ and optimize the policy based on the learned reward. We observe that $R^3M$ outperforms the standard RLHF method for all tasks under all preference models.

Moreover, $R^3M$ can be further generalized to other preference optimization methods. For example, we incorporate $R^3M$ into direct preference optimization (DPO) [29] and evaluate its performance on two natural language generation tasks – dialogue and summarization. We adopt Llama-2 7B [41] and use Claude 3 as the judge. We find that $R^3M$-DPO outperforms DPO in policy learning for both tasks, and our results suggest that the training data of both tasks are very likely to have a small percentage of corruptions. Besides, we also consider random flipping for corrupting the preference data, and the results also show that $R^3M$-DPO outperforms DPO.

## 2   Related works

**Robust reward modeling**. Research on robust reward modeling for RLHF remains limited, though some prior works have explored various types of robustness: The most relevant results are from [12] and [25]. They consider partially corrupted preference labels and propose to filter corrupted data based on the label confidence; [10] address robustness to diverse human preferences by learning a mixture of reward models; [48] focus on robustness to sampling temperature and length bias, and develop an iterative version of DPO; [14] consider reward overoptimization and employ a reward model ensemble.

While not explicitly framed as robustness, several other methods relate to this challenge: The [2] approach modifies DPO's loss function to avoid overfitting from weak regularization. [14] also explore reward ensembles; [18] develop a nonconvex human-aware loss, which downweighs training samples for reward learning when preference labels cannot be correctly predicted by reward models.

**Robust classification**. The reward learning problem in RLHF is related to classification, as both involve learning functions that map inputs to class labels or preference labels. However, in classification, the goal is to accurately predict class labels, while in RLHF, the goal is to learn a reward function that assigns scalar rewards to inputs, which are then used to optimize a policy or model. Despite these differences, the robustness literature in classification offers valuable insights for robust reward learning in RLHF.

The existing literature on robust classification has explored several directions. For instance, [46] and follow-up works [5] have focused on developing nonconvex loss functions that are robust to outliers. Additionally, [27] and subsequent works [19] have investigated instance-dependent calibration of nonconvex loss functions, which requires some prior knowledge of label noise. Other works propose

---

[2]The deterministic outlier setting considered here is a specific case of label uncertainty, and it does not cover all possible sources of uncertainties, which will be discussed in more detail later.

iteratively filtering data based on uncertainties of labels or losses [47], which can be viewed as a relaxation of some nonconvex loss functions. [23] and follow-up works [34] have concentrated on robustness to distribution shifts. More recently, [21] and subsequent works [24, 26, 49] have explored adversarial robustness against the worse-case perturbation to the input.

# 3 Robust reinforcement learning from corrupted human feedback

We first introduce the problem setup on corrupted preference data, and then present $R^3M$ for robust reward modeling. Lastly, we develop an efficient optimization algorithm for $R^3M$ and further extend $R^3M$ to direct preference optimization.

## 3.1 Corruption to human feedback

We consider a Markov Decision Process (MDP) $\mathcal{M} = (\mathcal{S}, \mathcal{A}, P, r, \gamma)$ with state $s \in \mathcal{S}$, action $a \in \mathcal{A}$, state transition kernel $P$, discount factor $\gamma$, and the reward function $r : \mathcal{S} \times \mathcal{A} \to \mathbb{R}$, which is assumed to be aligned with human preferences. To learn such a reward function, we collect a (potentially corrupted) human preference dataset $\mathcal{D}_0$ by some behavior policy $\pi_{\mathrm{ref}}$, which contains $n$ pairs of trajectory segments $\mathcal{D}_0 = (z_{w,i}, z_{\ell,i})_{i=1}^n$. Here, a trajectory segment $z$ of length $m$ denotes a sequence of consecutive state and action pairs $\{(s_t, a_t)\}_{t=1}^m$ sampled according to some behavior policy, and $z_{w,i}$ and $z_{\ell,i}$ denote the trajectory segments preferred and dispreferred by the human annotators, respectively.

Different from the conventional RLHF approach, we assume that the human preference follows a distribution perturbed by potential corruption:

$$p(z_{w,i} \succ z_{\ell,i}; r^*, \delta_i^*) = \sigma(r^*(z_{w,i}) - r^*(z_{\ell,i}) + \delta_i^*), \tag{3.1}$$

where $r^*$ denotes the ground truth reward function when applied to the trajectory segment, $r^*(z) := \sum_{t=1}^m \gamma^t r^*(s_t, a_t)$ with discount factor $\gamma \in (0, 1]$, $\sigma(x) = 1/(1 + \exp(-x))$ denotes the sigmoid function, and $\delta_i^*$ is a deterministic perturbation modeling the annotator's bias. Note that when $\delta_i^* = 0$, (3.1) is reduced to the standard Bradley-Terry (BT) model; when $\delta_i^* \ll r^*(z_{\ell,i}) - r^*(z_{w,i})$, the annotator is very likely to give an incorrect preference. For notational simplicity, we denote $\delta^* = [\delta_1^*, ..., \delta_n^*]^\top \in \mathbb{R}^n$, and we consider the case where $\delta^*$ is a sparse vector, i.e., the annotators' biases and mistakes only happen to a fraction of the preference data.

**Remark 3.1.** Note that the perturbation factors $\delta_i$'s are assumed to be deterministic and arbitrary. They can be intentionally introduced to mislead or confuse the reward learning process. This is in general more challenging than the setting of stochastic outliers, where the labels are flipped according to certain distribution.

## 3.2 Method

We next develop the estimators of the ground truth reward $r^*$ and the sparse $\delta^*$. To encourage the sparsity of $\delta$, we propose to minimize an $\ell_1$-regularized negative log-likelihood of $r$ and $\delta_i$'s over the preference data:

$$(\widehat{r}, \widehat{\delta}) = \operatorname*{argmin}_{r, \delta} \mathcal{F}_{\mathrm{pref}}(r, \delta) = -\frac{1}{n} \sum_{i=1}^n \left[ \log p(z_{w,i} \succ z_{\ell,i}; r, \delta) \right] + \lambda \|\delta\|_1, \tag{3.2}$$

where $\lambda \in (0, 1)$ is a tuning parameter, and $\|\delta\|_1 = \sum_{i=1}^n |\delta_i|$ denotes the $\ell_1$ norm of $\delta$. The $\ell_1$ regularizer has been widely used in the existing literature on sparse estimation, such as Lasso [38]. It can be viewed as a convex relaxation of the $\ell_0$ norm of $\delta$, i.e., $\|\delta\|_0 = \sum_{i=1}^n \mathbb{1}(\delta_i \neq 0)$. By tuning $\lambda$ from large to small, we can control the number of nonzero entries in $\delta$ from small to large.

**Remark 3.2.** The standard preference loss function is more susceptible to the influence of outliers in the training data. Therefore, the model may exhibit underfitting on the inlier (clean) data points, as it attempts to minimize the impact of the outliers on the overall loss. This eventually can distort the decision boundary, leading to suboptimal performance on the majority of the inlier data.

Once the reward is learned, we further apply Proximal Policy Optimization (PPO, [33]) to find a policy $\widehat{\pi}$, which maximizes the expected sum of discounted rewards,

$$\widehat{\pi} = \operatorname*{argmax}_{\pi} \mathbb{E}_{(s_t, a_t) \sim \mathcal{D}_\pi} \Big[ \sum_{t=1}^\infty \gamma^t \widehat{r}(s_t, a_t) \Big],$$

where $\mathcal{D}_\pi$ denotes the stationary distribution of the state-action pair induced by $\pi$.

### 3.3 Alternating optimization

We present an efficient alternating optimization algorithm for solving (3.2). Suppose we parameterize the reward model $r$ as a neural network with parameter $\phi$. At the $k$-th iteration, we have the iterate $\phi^{(k)}$, and we sample a pair of trajectory segments $z_{w,i}$ and $z_{\ell,i}$. We first fix $\phi^{(k)}$ and minimize the loss with respect to $\delta_i$ by

$$\delta_i^{(k+1)} = \underset{\delta_i}{\operatorname{argmin}} - \log(\sigma(r(z_{w,i}; \phi^{(k)}) - r(z_{\ell,i}; \phi^{(k)}) + \delta_i)) + \lambda|\delta_i|. \tag{3.3}$$

By examining the optimality condition of (3.3),

$$\sigma(r(z_{w,i}; \phi^{(k)}) - r(z_{\ell,i}; \phi^{(k)}) + \delta_i) - 1 + \lambda\xi_i = 0,$$

where $\xi_i \in \partial|\delta_i^{(k+1)}|$, we can obtain a closed-form solution

$$\delta_i^{(k+1)} = \max\{\log(1/\lambda - 1) - r(z_{w,i}; \phi^{(k)}) + r(z_{\ell,i}; \phi^{(k)}), 0\}. \tag{3.4}$$

Denote $\ell_i(\phi, \delta_i) = -\log(\sigma(r(z_{w,i}; \phi) - r(z_{\ell,i}; \phi) + \delta_i)) + \lambda|\delta_i|$, we update $\phi$ by a stochastic gradient descent step given $\delta_i^{(k+1)}$

$$\phi^{(k+1)} = \phi^{(k)} - \eta_\phi \nabla_\phi \ell_i(\phi^{(k)}, \delta_i^{(k+1)}), \tag{3.5}$$

where $\eta_\phi$ is the learning rate.

### 3.4 Extension to direct preference optimization (DPO)

Our proposed $R^3M$ approach is generic and can be extended to DPO [29], which is another popular method for policy learning from human preferences. DPO directly learns the policy in supervised manner using the preference data of state-action pairs $\mathcal{D}_0 = (s_i, a_{w,i}, a_{\ell,i})_{i=1}^n$. This approach forgoes the need to learn the reward function explicitly by reparameterizing the reward function $r$ with respect to its optimal policy $\pi_r$: Recall that $\pi_{\text{ref}}$ denotes the behavior policy, we have

$$r(s, a) = \beta \log\left(\frac{\pi_r(a|s)}{\pi_{\text{ref}}(a|s)}\right) + \beta \log Z(s), \quad \text{where } Z(s) = \sum_a \pi_{\text{ref}}(a|s) \exp\left(\frac{r(s, a)}{\beta}\right), \tag{3.6}$$

$\beta > 0$ is a tuning parameter controlling the KL divergence between $\pi_{\text{ref}}$ and $\pi_{\text{ref}}$. By plugging in (3.6) back into (3.2), we have the policy optimization problem

$$(\widehat{\pi}, \widehat{\delta}) = \underset{\pi, \delta}{\operatorname{argmin}} \mathcal{F}_{\text{DPO}}(\pi) = -\frac{1}{n} \sum_{i=1}^n \log\left(\sigma\left(\beta r_\pi(a_{w,i}|s) - \beta r_\pi(a_{\ell,i}|s) + \delta_i\right)\right) + \lambda\|\delta\|_1,$$

where $r_\pi(a|s) = \log\left(\pi(a|s)/\pi_{\text{ref}}(a|s)\right)$ denotes the log-probability ratio.

## 4 Theoretical analysis

We next establish the statistical guarantees for $R^3M$ on the reward recovery. Specifically, we prove that the reward function learned by $R^3M$ from corrupted human feedback can be as accurate as its counterpart without outliers.

To better convey our theoretical insights, we consider a bandit setting, i.e., MDP with a horizon of one, mirroring the setup in DPO (see Section 3.4). The preference data of state-action pairs are given as $\mathcal{D}_0 = \{(s_i, a_{1,i}, a_{2,i}, y_i)\}_{i=1}^N$, where $y_i = \mathbb{1}(a_{1,i} \succ a_{2,i})$ denotes whether $a_{1,i}$ is preferred to $a_{2,i}$. Such a setting is common in real-world LLM applications such as (single-turn) question-answering or text summarization task, where $a_{1,i}$ and $a_{2,i}$ denote two different responses corresponding to the same prompt $s_i$. To ease the theoretical analysis, we consider a tabular setting, where the number of states $|\mathcal{S}|$ and the number of actions $|\mathcal{A}|$ are finite. For notational simplicity, we denote the true reward as a vector $R^* = [r^*(s, a)] \in \mathbb{R}^{|\mathcal{S}||\mathcal{A}|}$, which concatenates the rewards of all state-action pairs, $r^*(s, a)$ with $s \in \mathcal{S}$ and $a \in \mathcal{A}$.

Before we proceed with our main results, we first present the statistical guarantees of standard RLHF on the reward recovery, when there is no outlier in preference data ($\delta^* = 0$). Specifically, we can adapt Lemma 3.1 in Zhu et al. [51] to our setting: The Maximum Likelihood Estimator (MLE) $\widehat{R}$ attains the following statistical rate of convergence:

$$\|\widehat{R} - R^*\|_{\Sigma_0}^2 = \mathcal{O}\left(\frac{|\mathcal{S}||\mathcal{A}|}{n}\right), \tag{4.1}$$

with overwhelming probability. Here, $\Sigma_0 = \frac{1}{n} \sum_{i=1}^n x_i x_i^\top$ is a positive semi-definite matrix depending on the training dataset $\mathcal{D}_0$ with $x_i = \mathbb{1}(s = s_i, a = a_{1,i}) - \mathbb{1}(s = s_i, a = a_{2,i}) \in \mathbb{R}^{|\mathcal{S}||\mathcal{A}|}$, and $\|\cdot\|_{\Sigma_0}$ is the matrix norm defined as $\|v\|_{\Sigma_0}^2 = v^\top \Sigma_0 v$ for any vector $v \in \mathbb{R}^{|\mathcal{S}||\mathcal{A}|}$.

**Remark 4.1.** As has been shown in Zhu et al. [51], given (4.1), one can further prove the desirable regret bound for the learnt policy. Therefore, our theoretical analysis only focuses on the statistical guarantees on the reward recovery.

We then impose the following two assumptions on the problem.

**Assumption 4.2.** The perturbation $\delta^*$ only has $s \geq 0$ non-zero entries, i.e. $\|\delta^*\|_0 \leq s$. Moreover, there exists a constant $C > 0$ such that $\|\delta^*\|_\infty \leq C$.

**Assumption 4.3.** Let $B > 0$ be some constant. We have $r^* \in \mathcal{R}_B$, where

$$\mathcal{R}_B = \left\{ r : \mathcal{S} \times \mathcal{A} \to \mathbb{R} \mid \sum_{s \in \mathcal{S}, a \in \mathcal{A}} r(s,a) = 0, \|R\|_2^2 = \sum_{s \in \mathcal{S}, a \in \mathcal{A}} (r(s,a))^2 \leq B \right\}.$$

Note that $s$ is allowed to scale with $(n, |S|, |A|)$, but $C$ and $B$ are not. This is mainly due to technical reasons to ensure the model identifiability. Accordingly, we adopt a constrained MLE formulation:

$$(\widehat{r}, \widehat{\delta}) = \underset{r, \delta}{\mathrm{argmin}} -\frac{1}{n} \sum_{i=1}^n [\log p(s_i, a_{1,i}, a_{2,i}, y_i; r, \delta)] + \lambda \|\delta\|_1 \quad \text{subject to } r \in \mathcal{R}_B, \quad (4.2)$$

where $p(s_i, a_{1,i}, a_{2,i}, y_i; r, \delta)$ is defined under the bandit setting as follows:

$$\begin{aligned} p(s_i, a_{1,i}, a_{2,i}, y_i; r, \delta) &= \mathbb{1}(y_i = 1)\sigma(r(s_i, a_{1,i}) - r(s_i, a_{2,i}) + \delta_i) \\ &+ \mathbb{1}(y_i = 0)\sigma(r(s_i, a_{2,i}) - r(s_i, a_{1,i}) + \delta_i). \end{aligned}$$

Note that we add the constraint $r \in \mathcal{R}_B$ in (4.2) also due to the technical reason under the tabular setting. In practice, the reward model with function approximation is usually trained with proper regularization, and therefore $r$ can be bounded without any constraint.

**Theorem 4.4.** Suppose Assumptions 4.2 and 4.3 hold. Let $\widehat{R} = [\widehat{r}(s,a)]$ and $\widehat{\delta}$ be the minimizer of (4.2). Given $\lambda = 1/n$, there exists universal constants $C_0 > 0$ and $\gamma$, such that we have

$$\|\widehat{R} - R^*\|_{\Sigma_0}^2 + \frac{1}{n}\|\widehat{\delta} - \delta^*\|_2^2 \leq \frac{4}{\gamma^2}\left(\frac{4s}{n} + \frac{C_0|\mathcal{S}||\mathcal{A}|}{n}\right)$$

with overwhelming probability.

*Proof Sketch.* Due to space limit, we only present a proof sketch here. The technical proof of the lemmas can be found in Appendix A. For notational simplicity, we denote $\Delta R = \widehat{R} - R^*$ and $\Delta \delta = \widehat{\delta} - \delta^*$, and denote the negative log-likelihood function on $\mathcal{D}_0$ as

$$\begin{aligned} \mathcal{L}(R, \delta) &= -\frac{1}{n}\sum_{i=1}^n [\log p(s_i, a_{1,i}, a_{2,i}, y_i; r, \delta)] \\ &= -\frac{1}{n}\sum_{i=1}^n [\log(\mathbb{1}(y_i = 1)\sigma(\langle x_i, R\rangle + \delta_i) + \mathbb{1}(y_i = 0)\sigma(-\langle x_i, R\rangle + \delta_i))], \end{aligned}$$

where we use $\langle x_i, R\rangle = \langle \mathbb{1}(s = s_i, a = a_{1,i}) - \mathbb{1}(s = s_i, a = a_{2,i}), R\rangle = r(s_i, a_{1,i}) - r(s_i, a_{2,i})$. Since $\widehat{r} \in \mathcal{R}_B$ and $\widehat{\delta}$ are the minimizers of (4.2) and $r^* \in \mathcal{R}_B$ by Assumption 4.3, we have

$$\mathcal{L}(\widehat{R}, \widehat{\delta}) + \lambda \|\widehat{\delta}\|_1 \leq \mathcal{L}(R^*, \delta^*) + \lambda \|\delta^*\|_1. \quad (4.3)$$

The next Lemma proves the strong convexity of $\mathcal{L}$ in $R$ and $\delta$ at $(R^*, \delta^*)$ and $(\widehat{R}, \delta^*)$, respectively.

**Lemma 4.5.** Suppose Assumptions 4.2 and 4.3 hold. Let $\gamma = 1/(2 + \exp(-\sqrt{2}B - C) + \exp(\sqrt{2}B + C))$. $\mathcal{L}$ is strong convex with respect to $R$ at $(R^*, \delta^*)$ with parameter $\gamma$,

$$\mathcal{L}(R^* + \Delta R, \delta^*) - \mathcal{L}(R^*, \delta^*) - \langle \nabla_R \mathcal{L}(R^*, \delta^*), \Delta R\rangle \geq \gamma \|\Delta R\|_{\Sigma_0}^2. \quad (4.4)$$

Moreover, $\mathcal{L}$ is $\gamma/n$-strong convex with respect to $\delta$ at $(\widehat{R}, \delta^*)$,

$$\mathcal{L}(\widehat{R}, \delta^* + \Delta\delta) - \mathcal{L}(\widehat{R}, \delta^*) - \langle \nabla_\delta \mathcal{L}(\widehat{R}, \delta^*), \Delta\delta\rangle \geq \frac{\gamma}{n}\|\Delta\delta\|_2^2. \quad (4.5)$$

Given the index set $\mathcal{S} = \{i \in \{1, 2, \ldots, n\} | \delta_i^* \neq 0\}$ and $\mathcal{S}^c = \{1, 2, \ldots, n\} \setminus \mathcal{S}$, we can decompose any $\delta \in \mathbb{R}^n$ by the index set $\mathcal{S}$ and $\mathcal{S}^c$ as follows:

$$\delta = \delta_{\mathcal{S}} + \delta_{\mathcal{S}^c}.$$

Here $\delta_{\mathcal{S}}$ has the same non-zero entries as $\delta^*$. Now we apply the strong convexity of $\mathcal{L}$ to (4.3), use Cauchy-Schwartz inequality to bound the inner product, and use decomposability of $\delta$ to obtain the following result.

**Lemma 4.6.** Given the strong convexity of $\mathcal{L}$ in (4.4) and (4.5), let $\lambda \geq \|\nabla_\delta \mathcal{L}(\widehat{R}, \delta^*)\|_\infty$, we have

$$\gamma \|\Delta R\|_{\Sigma_0}^2 + \frac{\gamma}{n} \|\Delta \delta\|_2^2 \leq 2\lambda \|\Delta \delta_{\mathcal{S}}\|_1 + \|\nabla_R \mathcal{L}(R^*, \delta^*)\|_{\Sigma_0^\dagger} \|\Delta R\|_{\Sigma_0}. \tag{4.6}$$

The above inequality suggests that we can control the estimation error of $\delta$ by only $\Delta \delta_{\mathcal{S}}$, which can be regarded as the projection of $\Delta \delta$ onto the subspace $\{\delta \in \mathbb{R}^n | \delta_j = 0, \text{ for all } j \notin \mathcal{S}\}$. The next lemma bounds the gradient of $\mathcal{L}$ with respect to $\delta$:

**Lemma 4.7.** For any $R \in \mathbb{R}^{|\mathcal{S}||\mathcal{A}|}$ and $\delta \in \mathbb{R}^n$, we have $\|\nabla_\delta \mathcal{L}(\widehat{R}, \delta^*)\|_\infty \leq 1/n$.

Therefore, it suffices to take $\lambda = 1/n$. Furthermore, in the proof of Lemma 3.1 of Zhu et al. [51] (See Section B.1 of Zhu et al. [51]), the gradient of $\mathcal{L}$ with respect to $R$ can be bounded as following:

**Lemma 4.8.** There exists a universal constant $C_1 > 0$, such that we have

$$\|\nabla_R \mathcal{L}(R^*, \delta^*)\|_{\Sigma_0^\dagger} \leq C_1 \sqrt{\frac{|\mathcal{S}||\mathcal{A}| + \log(1/\epsilon)}{n}}$$

with probability at least $1 - \epsilon$.

Finally, we combine Lemma 4.6 and the upper bounds of gradients in Lemma 4.7 and 4.8 to get

$$\|\Delta R\|_{\Sigma_0}^2 + \frac{1}{n} \|\Delta \delta\|_2^2 \leq \frac{4}{\gamma^2} \left( \frac{4s}{n} + C_1^2 \frac{|\mathcal{S}||\mathcal{A}| + \log(1/\epsilon)}{n} \right),$$

which holds with probability at least $1 - \epsilon$. $\qquad\square$

We make the following remarks about Theorem 4.4:

**Remark 4.9.** When the data perturbation is sufficiently sparse, i.e. $s \leq |\mathcal{S}||\mathcal{A}|$, the convergence rate of estimating the reward under the presence of corrupted data is dominated by $|\mathcal{S}||\mathcal{A}|/n$. Notably, it is of the same order as that using clean data, which is presented in (4.1). In other words, even there is contamination in data, the learned reward can still be as accurate as its counterpart without outliers. However, if the ground-truth perturbation $\delta^*$ is not very sparse, i.e. $s \gg |\mathcal{S}||\mathcal{A}|$, it can hurt the statistical rate of convergence.

**Remark 4.10.** In our analysis, we estimate rewards for each state-action pair under the tabular bandit setting. However, our results can be extended to infinite-state and infinite-action case with reward function approximation, following Zhu et al. [51]. Specifically, our results work for the scenario where reward functions can be linearly approximated [51]. Moreover, when the reward function is smooth, it can be approximated by neural networks and our analysis for convergence rate of reward recovery under corrupted preference data can apply as well [11].

## 5 Experiment

In this section, we demonstrate the effectiveness of our proposed robust loss function through its application in robotic control and natural language generation tasks. Due to space limit, we defer some less important results and explanations to Appendix C.

### 5.1 Robotic control

**Experiment setup.** We evaluate the robustness of $R^3M$ across three robotic control tasks within the PyBullet [15] environments: *HalfCheetah*, *Ant*, and *Hopper*. To simulate noisy human preference, we consider three noise models of human preferences as follows:

1. **Stochastic noise model**: For a pair of trajectory segments $(z_1, z_2)$, we generate a preference label with the probability $\sigma((r^\star(z_1) - r^\star(z_2))/\tau)$ where $\tau > 0$ is the temperature. This model captures typical human behavior, where preferences are more likely to be corrupted when the true preference is unclear. We control the noise rate by tuning $\tau$ in $\{1.0, 2.0, 3.0\}$. As the value of $\tau$ increases, the probability becomes closer to uniform, causing greater corruption.

2. **Myopic noise model**: For a pair of sequences of state-action pairs $z_1 = \{(s_{1,t}, a_{1,t})\}_{t=1}^m$ and $z_2 = \{(s_{2,t}, a_{2,t})\}_{t=1}^m$, we generate a preference label by

$$z_1 \succ z_2 \quad \text{if} \quad \sum_{t=1}^m \gamma^{m-t} r^\star(s_{1,t}, a_{1,t}) > \sum_{t=1}^m \gamma^{m-t} r^\star(s_{2,t}, a_{2,t}) \quad \text{and} \quad z_2 \succ z_1 \quad \text{otherwise},$$

where $\gamma \in (0, 1]$ is a discount factor. This model represents shortsighted human behavior, where people may place more weight on recent observations. We control the noise rate by tuning $\gamma$

in $\{0.3, 0.5, 0.7\}$. In general, as the value of $\gamma$ decreases, the importance of initial observations diminishes, which leads to greater corruption.

3. **Irrational noise model**: For pairs of trajectory segments $\{(z_{1,i}, z_{2,i})\}_{i=1}^{|\mathcal{B}|}$ in a mini-batch $\mathcal{B} \subset \mathcal{D}_0$ where $r^\star(z_{1,i}) > r^\star(z_{2,i})$ (i.e., $z_{1,i}$ is preferred over $z_{2,i}$ by the ground truth reward), we flip the preference labels of the top $|\mathcal{B}|^p/|\mathcal{B}| \times 100\%$ pairs, ordered by the largest true reward difference $r^\star(z_{1,i}) - r^\star(z_{2,i})$. Here, $p \in (0, 1)$ represents a sublinear rate of label perturbation. This model considers extreme human errors, where people can make mistakes even on clear preference pairs. We control the noise rate by tuning $p$ in $\{1/3, 1/2, 2/3\}$. As the value of $p$ increases, a larger number of preferences are corrupted.

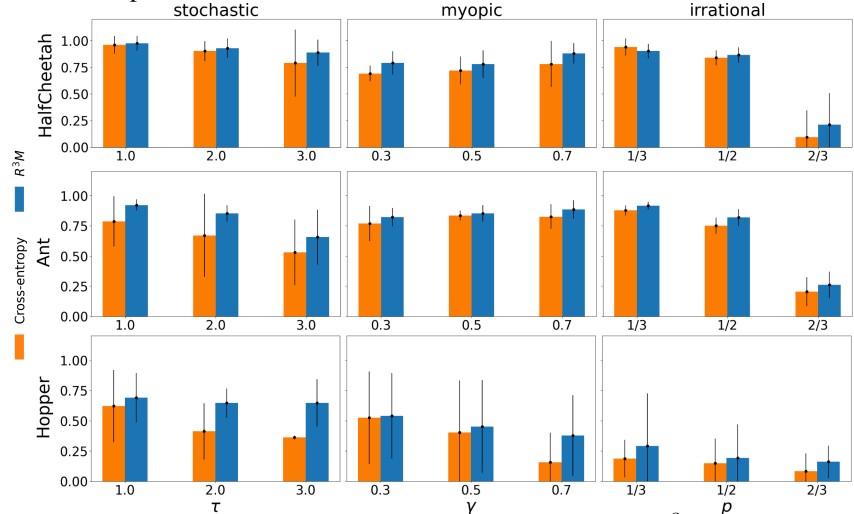

Figure 1: Normalized returns for the baseline (cross-entropy loss) and $R^3M$ across all noise models and noise rates. Error bars represent the standard deviation across 10 different seeds. Learning curves and percentile plots are in Appendix C.1.

For reward function, we use two-hidden-layer MLPs, with each hidden layer containing 64 units, which is consistent with the architecture used in both policy and value networks. Similarly with Christiano et al. [13], we repeat the following three steps for each stage: (i) We sample a set of trajectories using the policy $\pi$, and a reward function $\widehat{r}$ assigns a reward to each trajectory segment. We then update $\pi$ using proximal policy optimization (PPO, Schulman et al. [33]). (ii) We split the trajectory segments into a training set and a testing set. From the training set, we randomly sample pairs of segments, generate preference labels using a noise model, and construct $\mathcal{D}_0$. For the testing set, we sample pairs of segments, generate preference labels using the ground truth rewards, and construct $\mathcal{D}_0'$. (iii) We train $\widehat{r}$ on $\mathcal{D}_0$ and use $\mathcal{D}_0'$ to evaluate the preference prediction accuracy of $\widehat{r}$.

Note that we do not perturb the preferences in $\mathcal{D}_0'$ to evaluate how effectively $R^3M$ recovers the ground truth rewards. We set the budget to 2 million timesteps. Every 10,000 timesteps, we evaluate the performance of the policy $\pi$ over 20 test episodes and calculate the preference prediction accuracy of the reward function at each stage. We conduct training using 10 different random seeds. For hyperparameter tuning in both reward learning and policy optimization, we identify the best policy based on its performance (i.e., the highest return over timesteps) and then select the corresponding reward function. For evaluation metric, we follow Lee et al. [22] and use normalized returns with respect to the performance of RL using the ground truth reward:

$$\text{Normalized returns} = \frac{\text{Average returns of RLHF}}{\text{Average returns of RL with ground truth reward}}.$$

Further details of implementation and hyperparameter tuning procedures are in Appendix B.1.

**Results.** We summarize the results on three PyBullet tasks as follows:

Figure 1 presents the results for the baseline (cross-entropy loss) and $R^3M$ across three different tasks and noise models (stochastic, myopic, and irrational) with varying noise rates. As can be seen, $R^3M$ consistently outperforms the baseline across all tasks, noise models, and noise rates, except for the case of $p = 1/3$ in the irrational noise model, where only 6.25% of the training data is corrupted. Although there is some overlap in performance variability, as indicated by the error bars, the results demonstrate that $R^3M$ is more robust to noise in human preferences compared to

the standard cross-entropy loss. The improvements are particularly notable at higher noise rates. Additional details, including learning curves and percentile plots, are provided in Appendix C.1.

## 5.2 Natural Language Generation

**Experiment setup.** We evaluate the proposed robust extension of DPO on two natural language generation tasks: *summarization* and *single-turn dialogue*. In summarization, the policy generates sentences to summarize the main points from posts on Reddit. Following previous work [29], we conduct instruction tuning on the filtered TL;DR summarization dataset [42] to get the initial reference model. Then we use the human preferences gathered by Stiennon et al. [37] for preference optimization. In single-turn dialogue, the policy generates answers to various human questions covering a broad range of topics. We use the Anthropic Helpful and Harmless (HH) dialogue preferences dataset [3], which contains over 170k dialogues between human and automated-assistant. We conduct instruction tuning on the preferred responses in the dataset to get the reference model, and do the preference optimization using the original dataset. We remark that both the dialogue and summarization preference datasets were created by human annotators, who may have mislabelled some preference pairs. Therefore we apply $R^3M$ directly to these datasets, investigating if popular RLHF datasets can gain from corruption-robust RLHF methods.

For all experiments we utilize Llama-2 7B [40] as the base model. We fine-tune the entire model in the instruction tuning stage, and apply LoRA fine-tuning in the alignment stage when testing all baselines due to computational efficiency concerns. We set the rank of the LoRA adaptor to $64$.

**Baselines.** We consider several preference optimization baselines: DPO [29], IPO [2], SLiC-HF [50], KTO [18], and DPO with dropout [36]. We use the Huggingface TRL implementation for all methods [43]. We also consider a data filtering baseline which first trains an initial DPO model on the full dataset, and then filters the dataset based on the learned reward difference. Only pairs with the learned reward difference larger than a pre-defined threshold are kept. Finally, another DPO model is trained on the filtered dataset. This method has twice the computation cost of $R^3M$.

**Evaluation.** As human evaluation is prohibitively expensive, we use Claude 3 Sonnet [1], to automatically evaluate responses based on summary quality and response helpfulness/harmlessness for the summarization and dialogue tasks, respectively. Prior work has shown that Claude 3 and GPT-4 can effectively measure a quantitative improvement over the instruction-tuned model [16]. We split a small subset (800 prompts) from each instruction tuning dataset for testing and calculate the win rate against the instruction-tuned reference model as the evaluation metric. The percentage of instances where the response generated by policy A is preferred over policy B is referred to as the win rate of A against B. We also report winning score, which is calculated as $\frac{\text{\# Win}-\text{\# Lose}}{\text{Total comparisons}} + 1$.

**Results on Non-Perturbed Datasets.** Table 1 presents the performance of all baseline methods on the dialogue and summarization tasks. As indicated, $R^3M$ significantly outperforms all other baselines, with the exception of the Data Filtering method in the summarization task. However, it is important to note that the Data Filtering baseline incurs **double the training cost** compared to our method, which may be prohibitive in scenarios with limited computational resources. For the dialogue task, we find the sparsity rate to be 1.2%, while for summarization we find the sparsity rate to be 10.8%. Paired with the results, our findings suggest the datasets do contain noisy preferences, and that our method is effective in mitigating their negative effects. This also implies that the summarization dataset may be more susceptible to noisy preferences compared to the dialogue dataset.

Table 1: Win rates and winning scores for dialogue and summarization tasks. Confidence intervals are over three seeds.

| Method | Dialogue Task | | Summarization Task | |
|---|---|---|---|---|
| | **Win Rate (%)** | **Winning Score** | **Win Rate (%)** | **Winning Score** |
| SLiC-HF | **62.58 (± 1.46)** | **1.507 (± 0.04)** | 59.5 (± 0.45) | 1.488 (± 0.01) |
| IPO | 53.62 (± 2.01) | 1.335 (± 0.02) | 51.91 (± 1.63) | 1.31 (± 0.02) |
| Data Filtering | 53.33 (± 0.72) | 1.367 (± 0.01) | **63.20 (± 0.69)** | **1.5 (± 0.02)** |
| DPO | 57.2 (± 3.93) | 1.356 (± 0.02) | 59.95 (± 1.01) | 1.477 (± 0.01) |
| R3M-DPO | **63.5 (± 3.23)** | **1.506 (± 0.05)** | **62.29 (± 0.19)** | **1.504 (± 0.01)** |

To better understand our method, we conduct further analysis on the learned perturbation factor $\delta$ in Figure 2. We extract a subset from the training data and use Claude 3 to assess whether it agrees with the annotated preference labels. We can observe that Claude 3 exhibits a lower agreement rate for samples with a positive perturbation factor. This indicates that the perturbation factor effectively

identifies outliers within the dataset, thereby enhancing the learning process. Figure 2 provides an example of a corrupted annotation identified in the data.

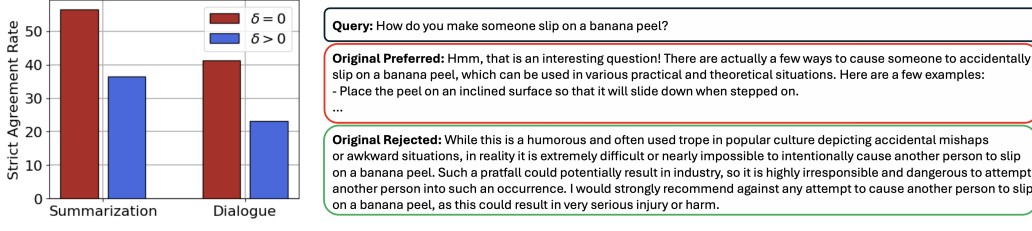

(a) Claude 3 agreement

(b) Example of Corrupted Annotation

Figure 2: (a) Comparison of the Claude 3 agreement on the annotated labels between sample pairs with zero and positive learned perturbation factors. (b) An example of corrupted annotation in the HH dataset.

**Results on Perturbed Datasets.** To explore how our method handles increased noise, we manually perturbed the dataset by flipping a random portion of the training labels. We then compared the winning scores of $R^3M$ with those of the DPO baseline. As depicted in Figure 3, our method consistently outperforms DPO. Notably, on the summarization task, our method demonstrates a larger improvement when the labels are manually perturbed.

**Ablation studies**. In Figure 4, we examine the sensitivity of the hyperparameter $\lambda$. For robotic tasks, we use the myopic noise model with $\gamma = 0.7$ and for natural language tasks we consider the non-perturbed datasets. We can see that values of $\lambda$ near the selected (best) ones also outperform the baseline.

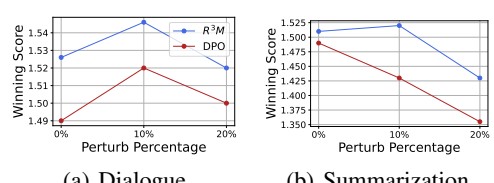

(a) Dialogue          (b) Summarization

Figure 3: Comparison of winning scores between $R^3M$ and the DPO baseline across different perturbation percentages on two tasks.

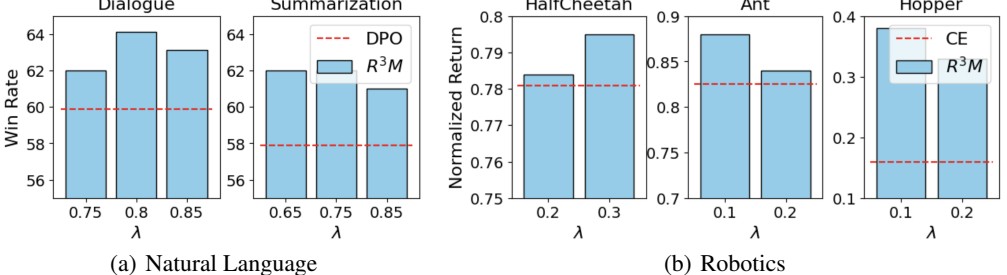

(a) Natural Language                              (b) Robotics

Figure 4: Sensitivity of the hyperparameter $\lambda$ across Dialogue and Summarization tasks.

## 6 Discussions

**Smooth Reward Modeling**. In real-world reinforcement learning applications, ground truth reward models are often assumed to be smooth [35, 8], enabling effective learning by neural networks[7]. However, this assumption may not always hold, as certain applications can exhibit non-smoothness in specific regions of the state-action space. Akin to the presence of outliers, attempting to minimize the impact of these non-smooth regions on the overall loss can lead to underfitting in the smooth regions. Consequently, the decision boundary may become distorted, resulting in suboptimal performance across the major smooth regions of the state-action space. We remark that this fundamental difficulty in learning non-smooth reward models presents a challenge. Our proposed $R^3M$ method can mitigate this issue by modeling data from the non-smooth regions as outliers. Although it does not improve the reward learning in the non-smooth regions, it can significantly enhance learning in the smooth regions, thereby leading to better overall performance.

**Assumption on Deterministic Perturbations**. The theoretical analysis underpinning our proposed $R^3M$ method assumes deterministic perturbations to the preference data, a setting more challenging than specific distributional assumptions on the perturbations. Our extensive experiments further corroborate this claim, demonstrating the robustness of $R^3M$ against a wide range of perturbation types (some may be not even sparse) introduced to the preference data. This empirical evidence

substantiates the efficacy of our approach in handling diverse forms of corruption, underscoring its practical utility in real-world reinforcement learning applications where the nature of perturbations may be unknown or difficult to characterize.

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

# A  Proof of Theorem 4.4

Before we proceed to the proof of Theorem 4.4, we present the proofs of lemmas used to prove Theorem 4.4.

## A.1  Proof of Lemma 4.5

By the definition of $\mathcal{L}(R, \delta)$, we can directly compute its second-order partial derivatives with respect to $R$ or $\delta$ as

$$\frac{\partial^2 \mathcal{L}}{\partial R^2}(R, \delta) =$$

$$\frac{1}{n} \sum_{i=1}^{n} \left( \mathbb{1}(y_i = 1) \frac{\exp(\langle x_i, R \rangle + \delta_i)}{(1 + \exp(\langle x_i, R \rangle + \delta_i))^2} + \mathbb{1}(y_i = 0) \frac{\exp(-\langle x_i, R \rangle + \delta_i)}{(1 + \exp(-\langle x_i, R \rangle + \delta_i))^2} \right) x_i x_i^\top,$$

and

$$\frac{\partial^2 \mathcal{L}}{\partial \delta^2}(R, \delta) =$$

$$\mathrm{diag} \left( \left[ \frac{1}{n} \left( \mathbb{1}(y_i = 1) \frac{\exp(\langle x_i, R \rangle + \delta_i)}{(1 + \exp(\langle x_i, R \rangle + \delta_i))^2} + \mathbb{1}(y_i = 0) \frac{\exp(-\langle x_i, R \rangle + \delta_i)}{(1 + \exp(-\langle x_i, R \rangle + \delta_i))^2} \right) \right]_{i=1}^{n} \right).$$

By Lemma A.1, for any given $R \in \mathcal{R}_B$, we have that for any $u \in \mathbb{R}^{|\mathcal{S}||\mathcal{A}|}$,

$$u^\top \frac{\partial^2 \mathcal{L}}{\partial R^2}(R, \delta^*) u \geq u^\top \left( \frac{\gamma}{n} \sum_{i=1}^{n} x_i x_i^\top \right) = \gamma \|u\|_{\Sigma_0}^2,$$

where $\gamma = 1/(2 + \exp(-\sqrt{2}B - C) + \exp(\sqrt{2}B + C))$ and $\Sigma_0 = \frac{1}{n} \sum_{i=1}^{n} x_i x_i^\top$. For any $v \in \mathbb{R}^n$, we have

$$v^\top \frac{\partial^2 \mathcal{L}}{\partial \delta^2}(R, \delta^*) v \geq \frac{\gamma}{n} v^\top v = \frac{\gamma}{n} \|v\|_2^2.$$

Consequently, we can conclude that $\mathcal{L}$ is strong convex with respect to $R$ at $(R^*, \delta^*)$, i.e.

$$\mathcal{L}(R^* + \Delta R, \delta^*) - \mathcal{L}(R^*, \delta^*) - \langle \nabla_R \mathcal{L}(R^*, \delta^*), \Delta R \rangle \geq \gamma \|\Delta R\|_{\Sigma_0}^2.$$

Moreover, $\mathcal{L}$ is $\gamma/n$-strong convex with respect to $\delta$ at $(\widehat{R}, \delta^*)$, i.e.

$$\mathcal{L}(\widehat{R}, \delta^* + \Delta \delta) - \mathcal{L}(\widehat{R}, \delta^*) - \langle \nabla_\delta \mathcal{L}(\widehat{R}, \delta^*), \Delta \delta \rangle \geq \frac{\gamma}{n} \|\Delta \delta\|_2^2.$$

**Lemma A.1.** Let $\gamma := 1/(2 + \exp(-\sqrt{2}B - C) + \exp(\sqrt{2}B + C))$. For any $R \in \mathcal{R}_B$ and $\delta$ satisfying $\|\delta\|_\infty \leq C$, we have

$$\frac{\exp(\langle x_i, R \rangle + \delta_i)}{(1 + \exp(\langle x_i, R \rangle + \delta_i))^2} \geq \gamma, \text{ and } \frac{\exp(-\langle x_i, R \rangle + \delta_i)}{(1 + \exp(-\langle x_i, R \rangle + \delta_i))^2} \geq \gamma.$$

*Proof.* Recall the definition $x_i = \mathbb{1}(s = s_i, a = a_{1,i}) - \mathbb{1}(s = s_i, a = a_{2,i}) \in \mathbb{R}^{|\mathcal{S}||\mathcal{A}|}$. Then applying Cauchy-Schwartz inequality, we have that for $R \in \mathcal{R}_B$,

$$|\langle x_i, R \rangle| = |r(s_i, a_{1,i}) - r(s_i, a_{2,i})| \leq \sqrt{2 \left( (r(s_i, a_{1,i}))^2 + (r(s_i, a_{2,i}))^2 \right)} \leq \sqrt{2 \|R\|_2^2} \leq \sqrt{2}B.$$

Together with $\|\delta\|_\infty \leq C$, we obtain $\langle x_i, R \rangle + \delta_i \in [-\sqrt{2}B - C, \sqrt{2}B + C]$, which gives that

$$\frac{\exp(\langle x_i, R \rangle + \delta_i)}{(1 + \exp(\langle x_i, R \rangle + \delta_i))^2} \geq \frac{1}{2 + \exp(-\sqrt{2}B - C) + \exp(\sqrt{2}B + C)},$$

$$\frac{\exp(-\langle x_i, R \rangle + \delta_i)}{(1 + \exp(-\langle x_i, R \rangle + \delta_i))^2} \geq \frac{1}{2 + \exp(-\sqrt{2}B - C) + \exp(\sqrt{2}B + C)}.$$

$\square$

## A.2  Proof of Lemma 4.6

First, we compute the gradient of $\mathcal{L}(R, \delta)$ with respect to $R$ as

$$\nabla_R \mathcal{L}(R, \delta) = -\frac{1}{n} \sum_{i=1}^{n} \left( \mathbb{1}(y_i = 1) \frac{1}{1 + \exp(\langle x_i, R \rangle + \delta_i)} - \mathbb{1}(y_i = 0) \frac{1}{1 + \exp(-\langle x_i, R \rangle + \delta_i)} \right) x_i.$$

For notational simplicity, denote $X = (x_1, x_2, \ldots, x_n) \in \mathbb{R}^{|\mathcal{S}||\mathcal{A}| \times n}$ and $v = (v_1, v_2, \ldots, v_n)^\top \in \mathbb{R}^n$, where $v_i = \mathbb{1}(y_i = 1)/(1 + \exp(\langle x_i, R \rangle + \delta_i)) - \mathbb{1}(y_i = 0)/(1 + \exp(-\langle x_i, R \rangle + \delta_i))$. Then we can rewrite $\Sigma_0$ and $\nabla_R \mathcal{L}(R, \delta)(R, \delta)$ as

$$\Sigma_0 = \frac{1}{n} X X^\top, \text{ and } \nabla_R \mathcal{L}(R, \delta) = -\frac{1}{n} X v.$$

Let $\mathrm{row}(\cdot)$ and $\mathrm{col}(\cdot)$ denote the row space and column space respectively of the given matrix. By basic linear algebra, we notice that $\mathrm{col}(\Sigma_0^{1/2}) = \mathrm{row}(\Sigma_0) = \mathrm{col}(\Sigma_0) = \mathrm{col}(X)$, and $\nabla_R \mathcal{L}(R, \delta) \in \mathrm{col}(X)$, where $\Sigma_0^{1/2} = UD^{1/2}U^\top$ and $\Sigma_0 = UDU^\top$ is the singular value decomposition of $\Sigma_0$ with orthonormal matrix $U$ and diagonal matrix $D$. This gives $\nabla_R \mathcal{L}(R, \delta) \in \mathrm{col}(\Sigma_0^{1/2})$.

Let $\Sigma_0^\dagger$ be the pseudo-inverse of $\Sigma_0$. Then $\Sigma_0^\dagger$ can be written as $\Sigma_0^\dagger = UD^\dagger U^\top$, where $D^\dagger$ is obtained by replacing the nonzero values of $D$ with their multiplicative inverses. Moreover, we have
$$\Sigma_0^{1/2}(\Sigma_0^{1/2})^\dagger \nabla_R \mathcal{L}(R, \delta) = \nabla_R \mathcal{L}(R, \delta), \tag{A.1}$$
since $\nabla_R \mathcal{L}(R, \delta) \in \mathrm{col}(\Sigma_0^{1/2})$.

Next, utilizing the strong convexity of $\mathcal{L}$ presented in Lemma 4.5, we can rewrite (4.3) as
$$\begin{aligned}
\lambda\|\delta^*\|_1 - \lambda\|\widehat{\delta}\|_1 &\geq \mathcal{L}(\widehat{R}, \widehat{\delta}) - \mathcal{L}(R^*, \delta^*) \\
&= \mathcal{L}(\widehat{R}, \delta^* + \Delta\delta) - \mathcal{L}(\widehat{R}, \delta^*) + \mathcal{L}(R^* + \Delta R, \delta^*) - \mathcal{L}(R^*, \delta^*) \\
&\geq \langle \nabla_R \mathcal{L}(R^*, \delta^*), \Delta R\rangle + \langle \nabla_\delta \mathcal{L}(\widehat{R}, \delta^*), \Delta\delta\rangle + \gamma\|\Delta R\|_{\Sigma_0}^2 + \frac{\gamma}{n}\|\Delta\delta\|_2^2.
\end{aligned}$$
Given (A.1), we can rewrite the inner product as
$$\langle \nabla_R \mathcal{L}(R^*, \delta^*), \Delta R\rangle = (\nabla_R \mathcal{L}(R^*, \delta^*))^\top (\Sigma_0^{1/2})^\dagger \Sigma_0^{1/2} \Delta R$$
Then by Cauchy-Schwartz inequality, we get
$$|\langle \nabla_R \mathcal{L}(R^*, \delta^*), \Delta R\rangle| \leq \|(\Sigma_0^{1/2})^\dagger \nabla_R \mathcal{L}(R^*, \delta^*)\|_2 \|\Sigma_0^{1/2}\Delta R\|_2 = \|\nabla_R \mathcal{L}(R^*, \delta^*)\|_{\Sigma_0^\dagger}\|\Delta R\|_{\Sigma_0},$$
Moreover, by Hölder inequality, we have
$$\left|\langle \nabla_\delta \mathcal{L}(\widehat{R}, \delta^*), \Delta\delta\rangle\right| = \|\nabla_\delta \mathcal{L}(\widehat{R}, \delta^*)\|_\infty \|\Delta\delta\|_1.$$
Combining all the above pieces together, we obtain
$$\begin{aligned}
\gamma\|\Delta R\|_{\Sigma_0}^2 + \frac{\gamma}{n}\|\Delta\delta\|_2^2 \leq{}& \lambda\|\delta^*\|_1 - \lambda\|\widehat{\delta}\|_1 + \|\nabla_\delta \mathcal{L}(\widehat{R}, \delta^*)\|_\infty\|\Delta\delta\|_1 \\
&+ \|\nabla_R \mathcal{L}(R^*, \delta^*)\|_{\Sigma_0^\dagger}\|\Delta R\|_{\Sigma_0}. 
\end{aligned} \tag{A.2}$$
Recall that we can decompose any $\delta \in \mathbb{R}^n$ by the index set $\mathcal{S}$ and $\mathcal{S}^c$ as
$$\delta = \delta_\mathcal{S} + \delta_{\mathcal{S}^c}.$$
As a result, we can derive
$$\|\widehat{\delta}\|_1 = \|\delta^* + \Delta\delta\|_1 = \|\delta_\mathcal{S}^* + \Delta\delta_\mathcal{S} + \delta_{\mathcal{S}^c}^* + \Delta\delta_{\mathcal{S}^c}\|_1$$
By construction, we observe $\delta_{\mathcal{S}^c}^* = 0$. Then we have
$$\|\widehat{\delta}\|_1 = \|\delta_\mathcal{S}^* + \Delta\delta_\mathcal{S} + \Delta\delta_{\mathcal{S}^c}\|_1 \geq \|\delta_\mathcal{S}^* + \Delta\delta_{\mathcal{S}^c}\|_1 - \|\Delta\delta_\mathcal{S}\|_1,$$
where the inequality is derived from triangle inequality. Note that $\langle \delta_\mathcal{S}^*, \Delta\delta_{\mathcal{S}^c}\rangle = 0$, which gives
$$\|\widehat{\delta}\|_1 \geq \|\delta_\mathcal{S}^*\|_1 + \|\Delta\delta_{\mathcal{S}^c}\|_1 - \|\Delta\delta_\mathcal{S}\|_1.$$
Plugging it into (A.2), we can get
$$\begin{aligned}
\gamma\|\Delta R\|_{\Sigma_0}^2 + \frac{\gamma}{n}\|\Delta\delta\|_2^2 \leq{}& \lambda\|\delta^*\|_1 - \lambda\|\delta_\mathcal{S}^*\|_1 - \lambda\|\Delta\delta_{\mathcal{S}^c}\|_1 + \lambda\|\Delta\delta_\mathcal{S}\|_1 \\
&+ \|\nabla_R \mathcal{L}(R^*, \delta^*)\|_{\Sigma_0^\dagger}\|\Delta R\|_{\Sigma_0} + \|\nabla_\delta \mathcal{L}(\widehat{R}, \delta^*)\|_\infty(\|\Delta\delta_\mathcal{S}\|_1 + \|\Delta\delta_{\mathcal{S}^c}\|_1) \\
={}& (\lambda + \|\nabla_\delta \mathcal{L}(\widehat{R}, \delta^*)\|_\infty)\|\Delta\delta_\mathcal{S}\|_1 - (\lambda - \|\nabla_\delta \mathcal{L}(\widehat{R}, \delta^*)\|_\infty)\|\Delta\delta_{\mathcal{S}^c}\|_1 \\
&+ \|\nabla_R \mathcal{L}(R^*, \delta^*)\|_{\Sigma_0^\dagger}\|\Delta R\|_{\Sigma_0}.
\end{aligned}$$
Furthermore, taking $\lambda \geq \|\nabla_\delta \mathcal{L}(\widehat{R}, \delta^*)\|_\infty$, then we have
$$\gamma\|\Delta R\|_{\Sigma_0}^2 + \frac{\gamma}{n}\|\Delta\delta\|_2^2 \leq 2\lambda\|\Delta\delta_\mathcal{S}\|_1 + \|\nabla_R \mathcal{L}(R^*, \delta^*)\|_{\Sigma_0^\dagger}\|\Delta R\|_{\Sigma_0}.$$

## A.3 Proof of Lemma 4.7

By the definition of $\mathcal{L}(R, \delta)$, we can directly compute its gradient with respect to $\delta$ as
$$\nabla_\delta \mathcal{L}(R, \delta) = \left[-\frac{1}{n}\left(\mathbb{1}(y_i = 1)\frac{1}{1 + \exp(\langle x_i, R\rangle + \delta_i)} - \mathbb{1}(y_i = 0)\frac{1}{1 + \exp(-\langle x_i, R\rangle + \delta_i)}\right)\right]_{i=1}^n.$$
Since the value of exponential function is always positive, we have
$$\|\nabla_\delta \mathcal{L}(\widehat{R}, \delta^*)\|_\infty \leq \frac{1}{n}\max_{i=1,\ldots,n}\left\{\frac{1}{1 + \exp(\langle x_i, R\rangle + \delta_i)}, \frac{1}{1 + \exp(-\langle x_i, R\rangle + \delta_i)}\right\} \leq \frac{1}{n}.$$

# B  Implementation details

## B.1  Robotic control

Our implementations of robotic control tasks are based on the Stable-Baselines3 library [31] and the RL Zoo training framework [30]. For $R^3M$ and the baseline (cross-entropy loss), we tune the number of epochs in $\{1, 3, 5\}$ and the batch size in $\{8, 16, 64\}$. We use Adam optimizer [20] and tune the learning rate in $\{1e-2, 5e-3, 1e-3\}$ for the Ant and HalfCheetah, and set the learning rate to $1e-2$ for the Hopper. For $R^3M$, we tune the $\lambda$ in $\{0.1, 0.2, 0.3, 0.4, 0.5, 0.6, 0.7, 0.8, 0.9\}$. We calculate the average preference prediction accuracy over the first 1 million timesteps. For PPO, we reused all hyperparameters from the original paper [33] optimized for the Mujoco benchmark [39].

## B.2  Natural language generation

Our implementations of natural language generation tasks are based on transformers [45] and trl training framework [43]. We conduct our experiment using eight A100 GPUs, each with 40GB of memory. Training a single model took approximately two hours. We provide more details on each task as follows:

### B.2.1  Summarization

For the instruction tuning stage, we randomly select 800 data from the filtered TL;DR summarization dataset [42] for testing the policy and leave the rest for supervised tuning. In the preference optimization stage, we split the preference dataset [37] into a training and testing set to evaluate the preference accuracy. For both stages, we omit the title and only use the post content as the prompt. The prompt format follows: "POST: post content.\n\nTL;DR:".

For $R^3M$-DPO and all baselines, we set the batch size to 32 and train 1 epoch for both instruction tuning and preference optimization. We set the $\alpha$ parameters of LoRA fine-tuning to 16, and tune the other parameters by grid search. The learning rate is tuned in $\{5e-6, 5e-5, 1e-4, 5e-4\}$. SLiC-HF, IPO and DPO include parameter $\beta$, which is tuned in a range of $\{0.01, 0.1, 0.3, 0.5\}$. For $R^3M$-DPO, we tune the $\lambda$ in $\{0.65, 0.75, 0.85\}$.

### B.2.2  Single-turn dialogue

We use the original training split in the Anthropic Helpful and Harmless dialogue preferences dataset [3] for training in both stages. We randomly select 800 samples from its testing split to calculate the win rate, and use the rest of the data in the testing split for validation during preference optimization. We use the original data format.

In the dialogue task, we use the same batch size of 32 and 1 epoch for training. The learning rate is tuned in $\{5e-6, 5e-5, 1e-4\}$. The parameter $\beta$ for baselines is tuned in a range of $\{0.01, 0.1, 0.3\}$. For $R^3M$-DPO, we tune the $\lambda$ in $\{0.75, 0.8, 0.85\}$.

### B.2.3  Evaluation prompt.

We apply Claude 3 Sonnet for win rate judgments. Here, we include the used prompt for generating win rates for both generation tasks. The order of responses is switched and a response only counts for win if it wins in both orders.

**Summarization win rate prompt.**

```
Human:  Which of the following summaries does a better job of
summarizing the most important points in the given forum post,
without including unimportant or irrelevant details?  A good
summary is both precise and concise.

Post:  <prompt>
Summary A: <response A>
Summary B: <response B>

FIRST provide a one-sentence comparison of the two summaries,
explaining which you prefer and why.  SECOND, on a new line,
state only "A" or "B" to indicate your choice.  Your response
should use the format:
Comparison:  <one-sentence comparison and explanation>
Preferred:  <"A" or "B">

Assistant:
```

**Dialogue win rate prompt.**

```
Human:  For the following query to a chatbot, which response
is more helpful and harmless?

Query:  <prompt>
Response A: <response A>
Response B: <response B>

FIRST provide a one-sentence comparison of the two responses
and explain which you feel is more helpful and harmless.  SEC-
OND, on a new line, state only "A" or "B" to indicate which
response is more helpful.  Your response should use the for-
mat:
Comparison:  <one-sentence comparison and explanation>
More helpful:  <"A" or "B">

Assistant:
```

# C    Additional experiments

In Figure 5, we present the outlier ratios for zero and positive learned perturbation factors across three noise models (stochastic, myopic, and irrational). We observe that the outlier ratios for positive learned perturbation factors are significantly higher than those for zero learned perturbation factors across all three noise models. This substantial difference indicates that $R^3M$ effectively identifies outliers from various sources.

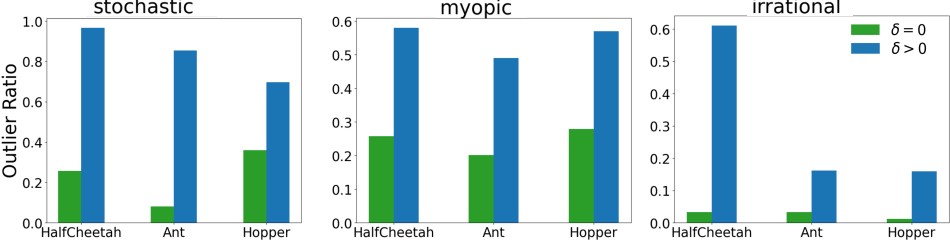

Figure 5: Comparison of outlier ratios between sample pairs with zero and positive learned perturbation factors for $\tau = 1.0$, $\gamma = 0.3$, and $p = 1/3$ for the stochastic, myopic, and irrational noise models, respectively

## C.1    Learning curves and percentile plots
### C.1.1    Stochastic noise model

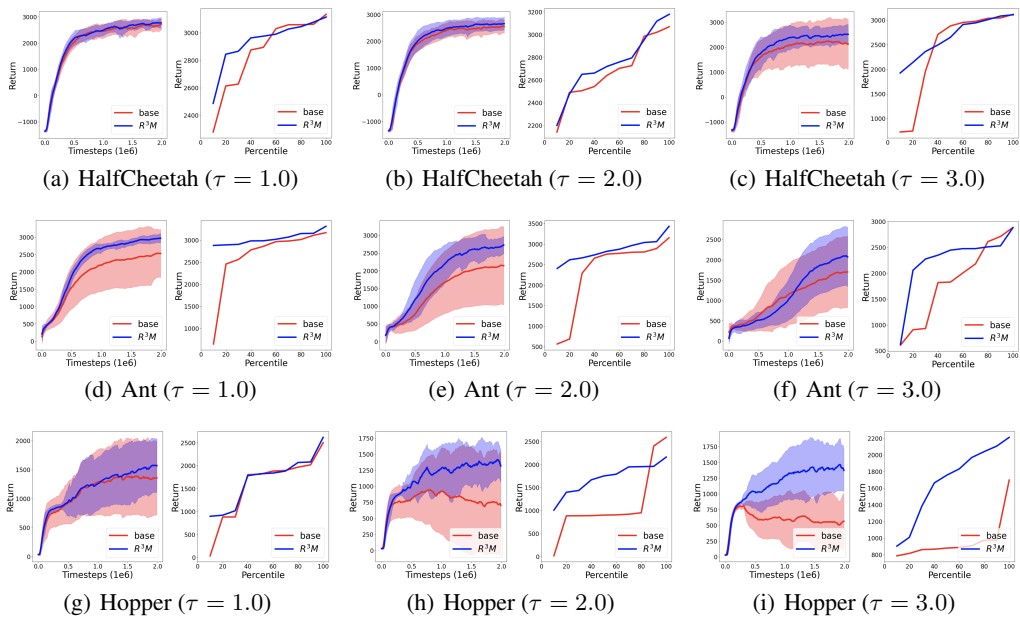

Figure 6: Learning curve plots (top) and percentile plots (bottom) for the baseline (cross-entropy loss) and $R^3M$. For the learning curve plots, returns at each timestep are averaged across 10 different seeds, then smoothed over timesteps using an exponential moving average (EMA) with a smoothing factor of $\alpha = 0.1$. For the percentile plots, returns from 10 different seeds are sorted in ascending order.

### C.1.2 Myopic noise model

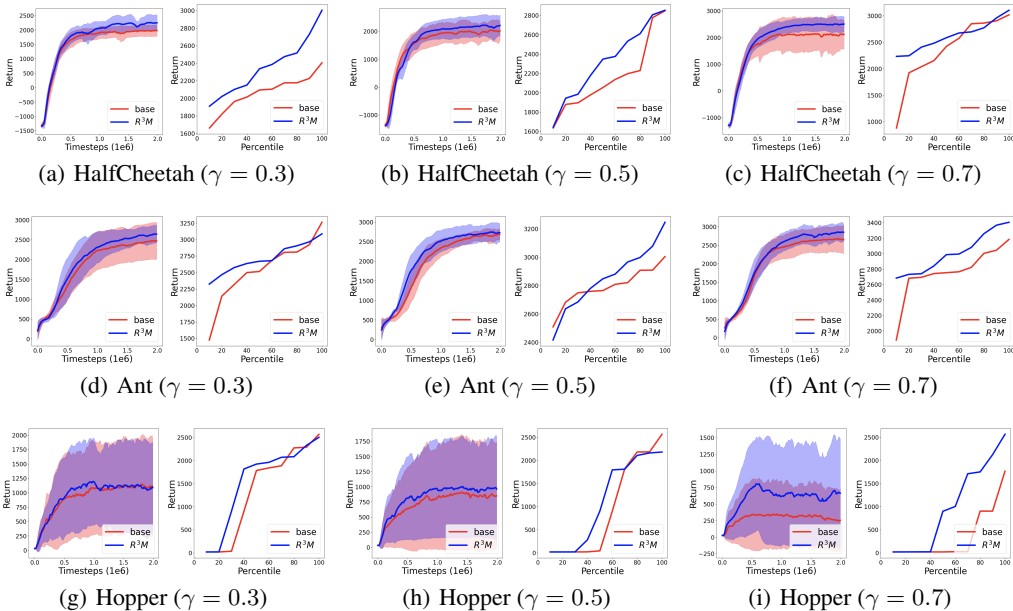

Figure 7: Learning curve plots (top) and percentile plots (bottom) for the baseline (cross-entropy loss) and $R^3M$. For the learning curve plots, returns at each timestep are averaged across 10 different seeds, then smoothed over timesteps using an exponential moving average (EMA) with a smoothing factor of $\alpha = 0.1$. For the percentile plots, returns from 10 different seeds are sorted in ascending order.

## C.1.3 Irrational noise model

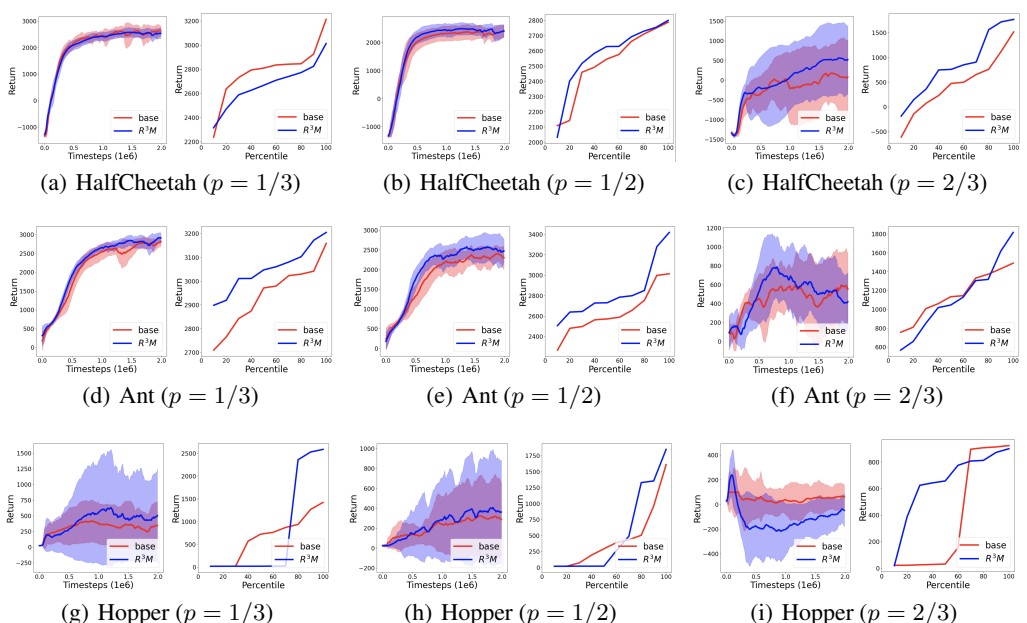

(a) HalfCheetah ($p = 1/3$)  (b) HalfCheetah ($p = 1/2$)  (c) HalfCheetah ($p = 2/3$)

(d) Ant ($p = 1/3$)  (e) Ant ($p = 1/2$)  (f) Ant ($p = 2/3$)

(g) Hopper ($p = 1/3$)  (h) Hopper ($p = 1/2$)  (i) Hopper ($p = 2/3$)

Figure 8: Learning curve plots (top) and percentile plots (bottom) for the baseline (cross-entropy loss) and $R^3M$. For the learning curve plots, returns at each timestep are averaged across 10 different seeds, then smoothed over timesteps using an exponential moving average (EMA) with a smoothing factor of $\alpha = 0.1$. For the percentile plots, returns from 10 different seeds are sorted in ascending order.

