# OpenReview forum: "Robust Reinforcement Learning from Corrupted Human Feedback"
_NeurIPS.cc/2024/Conference — NeurIPS 2024 poster_

### Official Review · Reviewer_ugMr · 2024-06-27

**Soundness:** 3
**Presentation:** 3
**Contribution:** 2
**Rating:** 5
**Confidence:** 4

**Summary:**

The paper presents a robust reward learning approach by formulating it as an $\ell_1$-regularized maximum likelihood estimation problem. And it also introduces an alternating optimization algorithm, which introduces minimal computational overhead when compared to the standard RLHF approach.

**Strengths:**

* The paper is well-written and contains two types of experiments: robotic control and text generation.
* The method is straightforward, and the motivation is reasonable.

**Weaknesses:**

* The paper lacks an experiment for formal RLHF, such as Proximal Policy Optimization (PPO), in the context of text generation. Including such an experiment could provide a more comprehensive evaluation of the proposed method.
* The use of $\delta$ as a regularization method may appear somewhat trivial for this approach. Conducting additional experiments on HH-RLHF or Ultrafeedback to provide further evidence and support for the effectiveness of this method would be beneficial.

**Questions:**

* It is unclear how far the Perturb Percentages can be extended before the model completely collapses. Further investigation and experimentation are needed to determine the limits and thresholds that lead to model failure.
* The difficulty level of optimizing $\delta" is not specified. It would be helpful to elaborate on the challenges associated with optimizing this parameter and any potential limitations encountered during the optimization process.
* In order to gain a comprehensive understanding of the effects and limitations of the proposed method, I hope the author can conduct experiments specifically focused on PPO.

**Limitations:**

The context lacks information regarding the limitations of the paper or the proposed method.  Besides, the analysis on DPO does not fully reflect the true impact of RM's signal on PPO, such as overrated scores. Hence, to adequately assess the effect of the real RLHF's RM on PPO, it is essential to conduct experiments specifically targeting PPO. The current approach lacks experimentation on PPO, which limits our understanding of how the RM's signal truly impacts this particular method

---

> ### Author Rebuttal · Authors · 2024-08-07
>
> We would like to thank you for your constructive comments! In the following, your comments are first started and then followed by our point-by-point responses.
>
> **W1, Q3, L1: The paper lacks an experiment for formal RLHF, such as Proximal Policy Optimization (PPO), in the context of text generation. Including such an experiment could provide a more comprehensive evaluation of the proposed method.**
>
> We agree that including experiments with PPO on NLP tasks can help fully comprehending the impact of our method. Please refer to point 4 in the global rebuttal.
>
> **W2: The use of $\delta$ as a regularization method may appear somewhat trivial for this approach. Conducting additional experiments on HH-RLHF or Ultrafeedback to provide further evidence and support for the effectiveness of this method would be beneficial.**
>
> Thanks for pointing this out! We conduct experiment on the Ultrafeedback dataset to further demonstrate the effectiveness of the proposed method. Results and details are included in point 1 in the global rebuttal.
>
> **Q1: It is unclear how far the Perturb Percentages can be extended before the model completely collapses. Further investigation and experimentation are needed to determine the limits and thresholds that lead to model failure.**
>
> Response is included in point 3 in the global rebuttal.
>
> **Q2: The difficulty level of optimizing $\delta$ is not specified. It would be helpful to elaborate on the challenges associated with optimizing this parameter and any potential limitations encountered during the optimization process.**
>
> We found that optimizing $\delta_i$ is not difficult, as $\delta_i$ has a closed-form solution in Eq. (3.4) for each iteration. Therefore, we can efficiently find the optimal $\delta_i$ without needing any optimization algorithms such as the proximal gradient descent  for solving Eq. (3.3).

---

> > ### Comment · Reviewer_ugMr · 2024-08-11
> >
> > Thank you for the authors' comprehensive response. I would like to maintain the current score and acknowledge the positive aspects. Thanks!

---

> > > ### Author Response · Authors · 2024-08-13
> > > **Thank you for your review**
> > >
> > > Dear Reviewer ugMr,
> > >
> > > We are grateful for your review and your positive evaluation of our work. Your suggestions are invaluable, and we will try to include the recommended experiments in the paper.

---

### Official Review · Reviewer_XAMr · 2024-07-12

**Soundness:** 3
**Presentation:** 3
**Contribution:** 3
**Rating:** 6
**Confidence:** 3

**Summary:**

The paper proposes a robust RLHF method which models the potentially corrupted preference label as sparse outliers. They prove that their method can consistently identify outliers in addition to learning the underlying reward functions, under proper conditions. The results on both robotic control and natural language generation tasks show that the proposed approach improves robustness of reward learning phase within RLHF framework.

**Strengths:**

This article raises an important issue in RLHF, that human annotators may give incorrect or inconsistent preference labels. and then introduces a simple but useful extension to the existing RLHF framework — including a perturbation factor. The authors theoretically prove the statistical rate of convergence in fitting the reward functions as well as the outliers, under particular conditions. They conduct thorough experiments over different noise modes of human preferences in robotic control, and further extend their approach to DPO within real-world language land to support their claims.

**Weaknesses:**

- Error bars are expected in Table 1.
- The Anthropic Helpful and Harmless dialogue preferences dataset has a high disagreement rate, so that flipping a random portion of the training labels may not increase noise as expected, which could probably cause the unexpected results. The authors should probably filter the dataset or try a cleaner dataset to show the results on perturbation.

**Questions:**

- What’s the relationship and difference between Win Rate and Winning Score in your Table 1?
- Could you try to draw a fair comparison between Data Filtering and $R^3M$-DPO? Although your Data Filtering baseline doubles the training cost, it still indicates that filtering the dataset could be as effective as modeling the outliers. One thing you could provide is the performance of $R^3M$-DPO over the filtered dataset.

**Limitations:**

The paper does not discuss the limitation of the work. Please discuss the potential issue of applying your approach to real-world RLHF tasks.

---

> ### Author Rebuttal · Authors · 2024-08-07
>
> **W1: Error bars are expected in Table 1.**
>
> Please refer to point 2 in the global rebuttal.
>
> **W2: The Anthropic Helpful and Harmless dialogue preferences dataset has a high disagreement rate, so that flipping a random portion of the training labels may not increase noise as expected, which could probably cause the unexpected results. The authors should probably filter the dataset or try a cleaner dataset to show the results on perturbation.**
>
> Since accurately filtering the dataset is challenging and there is no consensus on the best approach, we conducted the experiment on a cleaner, more modern dataset, Ultrafeedback. More details and analysis are included in Point 1 of the global rebuttal.
>
> **Q1: What’s the relationship and difference between Win Rate and Winning Score in your Table 1?**
>
> Win rate is defined as $\dfrac{\text{\\# Win}}{\text{\\# Total comparisons}}$ and Winning score is defined as $\dfrac{\text{\\# Win - \\# Lose}}{\text{\\# Total comparisons}}$ + 1. This means Winning Score additionally considers the "tie" cases compared to Win Rate. For example, two models with the same Win Rate can have different Winning Scores. The model with more tie cases would have a higher Winning Score. Therefore, we can view Win Rate as the primary criterion for evaluating models, and Winning Score as a secondary criterion that also accounts for tie cases.
>
> **Q2: Could you try to draw a fair comparison between Data Filtering and $R^3M$-DPO? Although your Data Filtering baseline doubles the training cost, it still indicates that filtering the dataset could be as effective as modeling the outliers. One thing you could provide is the performance of $R^3M$-DPO over the filtered dataset.**
>
> We believe that directly comparing Data Filtering with $R^3M$-DPO is fair and valid for noisy preference datasets. This is because our method improves the performance of standard DPO on noisy datasets, rather than on clean ones. With a noisy preference dataset, data filtering involves filtering the data first and then applying standard DPO to the filtered data. In contrast, our method does not require data filtering, as optimizing $\delta$ inherently performs a similar function.
>
>
> **L1: The paper does not discuss the limitation of the work. Please discuss the potential issue of applying your approach to real-world RLHF tasks.**
>
> Thank you for the suggestion! We provide our discussion as follows and will include it in our next version: Our approach does not introduce significant limitations compared to previous work. However, it does introduce an additional parameter, $\lambda$, in the regularization term, which may increase the tuning efforts in practical use. Developing more adaptive methods or conducting analysis on selecting this hyperparameter could be considered as future work.

---

> > ### Comment · Reviewer_XAMr · 2024-08-12
> >
> > Thank you for your response, and I appreciate the newly added results in global rebuttal. I would maintain the current score. Thanks!

---

> > > ### Author Response · Authors · 2024-08-13
> > > **Thank you for your review**
> > >
> > > Dear Reviewer XAMr,
> > >
> > > Thank you for your thorough review and for recommending our work for acceptance! Your insightful comments are invaluable in helping us further improve our paper.

---

### Official Review · Reviewer_m2Ct · 2024-07-13

**Soundness:** 2
**Presentation:** 2
**Contribution:** 2
**Rating:** 6
**Confidence:** 4

**Summary:**

The paper studies the problem of robust reinforcement learning when a small fraction of the human feedback preference data can be corrupted by adversary.

**Strengths:**

The paper formulates the robust RLHF problem and provides a straightforward and easy-to-use $\ell_1$ regularization algorithm for learning the reward signal. It provides experimental justification for the proposed algorithms.

**Weaknesses:**

I have a few questions regarding the formulation and experimental results. Please see next section.

**Questions:**

1. In practical dataset, how do we justify the correctness of sparse perturbation assumption rather than stochastic noise in the original Bradley-Terry model?

2. What is the criterion for tuning the parameter $\lambda$ in the regularized loss?

3. The TL;DR and HH dataset might not lead to significant improvement over preferences due to both responses being too much worse than current llama2 generation. Is it possible to run on modern preference dataset like HelpSteer2, Nectar or UltraFeedback, and report the Arena-Hard score as the evaluation?

---

> ### Author Rebuttal · Authors · 2024-08-07
>
> Thank you for your valuable feedback! We provide a detailed response to your questions as follows:
>
> **Q1: In practical dataset, how do we justify the correctness of sparse perturbation assumption rather than stochastic noise in the original Bradley-Terry model?**
>
> As discussed in Section 6 and Remark 3.1, the sparse perturbation assumption in the preference data is a more challenging setting compared to the stochastic noise assumption. Therefore, our method can also improve robustness against other types of noise. Our experimental results illustrate that our method performs better across a wide range of perturbation types, including stochastic noise. While it is challenging to accurately verify assumptions in practical datasets, our method demonstrates better results due to its robustness.
>
>
> **Q2: What is the criterion for tuning the parameter $\lambda$ in the regularized loss?**
>
> We generally tuned $\lambda$ by performing a grid search within the range (0, 1). For the tuning metric, we used different criteria depending on the task: for the robotic control task, we optimized $\lambda$ based on the true episode return, while for the natural language generation task, we optimized $\lambda$ based on the win rate against a supervised fine-tuned model.
>
>
>
> **Q3: The TL;DR and HH dataset might not lead to significant improvement over preferences due to both responses being too much worse than current llama2 generation. Is it possible to run on modern preference dataset like HelpSteer2, Nectar or UltraFeedback, and report the Arena-Hard score as the evaluation?**
>
> Response included in point 1 in the global rebuttal.

---

> > ### Comment · Reviewer_m2Ct · 2024-08-11
> > **Thanks**
> >
> > Thank you for your response. I appreciate the author’s efforts in running new experiments and most of my concerns are resolved. I have increased my score accordingly.
> >
> > One tiny suggestion is that llama2 might be too weak for arena hard. It might be better to start with at least llama 3 / 3.1 / Gemma 2. But I understand that during the short time window of rebuttal it can be very hard to finish the experiments. I’m excited to see the results in the future version of the paper!

---

> > > ### Author Response · Authors · 2024-08-11
> > > **Thank you for the discussion**
> > >
> > > Dear Reviewer m2Ct,
> > >
> > > Thank you for the engaging discussion and the willingness to raise your score! We will be sure to include the contents of the rebuttal in the paper, and we will try using our method with llama 3.1 and arena hard as well.

---

### Official Review · Reviewer_hEuH · 2024-07-15

**Soundness:** 3
**Presentation:** 3
**Contribution:** 3
**Rating:** 5
**Confidence:** 3

**Summary:**

This paper proposes a framework for robustifing learning from human preferences. It models noise and bias in human annotation of the dataset as an offset added to the true margin between the preferred and disprefered examples. It further utilizes L1 regularization to induce sparsity  in the offset. Empirically, the method is benchmarked on robotics and natural language through its extension to Direct Preference Optimization (DPO).

**Strengths:**

- The paper is generally well-written
- The method is well-motivated
- Empirical results are generally comprehensive, across two domains, and a hand-full of baselines are considered.

**Weaknesses:**

- The value of the $\delta$ offset could also be interpreted as by how much is the positive completion preferred over the rejected. So in a way, the method is modeling an explicit different in quality of the preference pairs. However, this offset may be orthogonal to label-noise e.g. examples that have small offset (positive and negative are close to one another) may be more difficult to annotate as there is not striking difference between the two completions. However, the statement on line 113
> ... the annotator is very likely to give an incorrect preference
only takes into account the opposite extreme.

Prior to this work, "Direct Preference Optimization with an Offset" explores DPO with an offset to model the difference in degree of preference (ODPO).

- Given that the canonical framework for RLHF of in LLMs is using REINFORCE or PPO with a learned reward model (through logistic regression) , results in that setting are necessary given the framing / positioning of the paper, where the application to DPO is a plesent bonus.  In that setting, the test-set classification accuracy of the reward model would of interest in the the different noise settings.

**Questions:**

- I am a bit concerned about the results in in Figure 3, from reading off the graph, the increase / decrease in Winning score when changing the perturb % is about the same for $R^3M$ and DPO for dialogue? It would be clearer if the authors simply plot the delta in the score as that's the metric underlying raw performance, that we care about in this context.

- More ablations along the extremes would help ground the experimental results further and increase the quality of the work. e.g. for $\lambda$, higher perturb % (100% perturbation would give a sense of how big the drops in performance are relative to their practical worst-case)

-Reporting the ranking accuracy when training DPO (i.e. % of time $r_w - r_l > 0$) with and without the robustification scheme would further strengthen the results.

**Limitations:**

No limitations have been provided.

---

> ### Author Rebuttal · Authors · 2024-08-07
>
> Thank you for your comprehensive review and valuable feedback. We provide a detailed response to your comments as follows:
>
> **W1: The value of the $\delta$ offset could also be interpreted as by how much is the positive completion preferred over the rejected. This offset may be orthogonal to label-noise. Prior to this work, "Direct Preference Optimization with an Offset" explores DPO with an offset to model the difference in degree of preference (ODPO).**
>
> Thank you for your discussion and suggestions regarding related literature. We would like to clarify that our parameter, $\delta$, is fundamentally different from the margin parameter, $\Delta$, used in ODPO [1]. Specifically:
>
> In our method, $\delta$ is jointly optimized with the reward model. By doing so, larger $\delta$ values are learned for corrupted preference samples to achieve smaller reward differences, compensating for the perturbations.
> In contrast, $\Delta$ in ODPO is not learnable but is a prefixed value proportional to the score difference between winning and losing responses. Optimizing the ODPO loss with larger $\Delta$ (i.e., pairs with strong preference strength) results in a larger learned difference, which is opposite to the effect of our $\delta$.
> When corrupted preferences (where the ranking of scores is flipped) are present, ODPO would exacerbate the label noise. Additionally, our $\delta$ parameter is sparse due to the use of an $\ell_1$ regularizer.
>
> We further illustrate the relationship between $\delta$ and the learned reward difference in the Ant task by Figure 1 in the attached file in global rebuttal. We categorized the learned reward differences into five percentile bins and then binned $\delta$ accordingly to compute the average for each bin. From the figure, it is evident that larger $\delta$ values correspond to smaller learned reward differences.
>
>
>
> **W2: Given that the canonical framework for RLHF of in LLMs is using REINFORCE or PPO with a learned reward model (through logistic regression) , results in that setting are necessary given the framing / positioning of the paper.**
>
> Response included in point 4 in the global rebuttal.
>
>
> **Q1: I am a bit concerned about the results in Figure 3, from reading off the graph, the increase / decrease in Winning score when changing the perturb \% is about the same for $R^3M$ and DPO for dialogue? It would be clearer if the authors simply plot the delta in the score as that's the metric underlying raw performance, that we care about in this context.**
>
> Thank you for your suggestion. Since both the HH dialogue and TL;DR summarization datasets contain over 20\% noise labels [3] and the response quality in these datasets is considered low, evaluating our methods solely on these datasets could lead to unexpected results, as you pointed out.
>
> To address this, we trained our method on the high-quality UltraFeedback dataset [3] with different levels of random perturbation. Due to resource limitations, we considered only 10\% and 20\% perturbations. The results are shown in Table 6 in the attached file. We evaluated the standard DPO and $R^3M$-DPO by their win rates against HH SFT. We define $\Delta$ as the win rate difference between two methods and present the value in the table. We will include the plot of delta in our next version.
>
> **Q2: More ablations along the extremes would help ground the experimental results further and increase the quality of the work. e.g. for $\lambda$, higher perturb \% (100\% perturbation would give a sense of how big the drops in performance are relative to their practical worst-case).**
>
> Response included in point 3 in the global rebuttal.
>
> **Q3: Reporting the ranking accuracy when training DPO (i.e. \% of time $r_w-r_l>0$) with and without the robustification scheme would further strengthen the results.**
>
> \textbf{Rebuttal: } Thank you for your suggestion. Including the ranking accuracy on the test set will provide additional support for our results. Table 5 in the attached file shows that training DPO with robustification scheme slightly improves the ranking accuracy compared to the one without robustification scheme. We would like to note that the accuracy can not completely reflect its quality towards policy optimization, since the accuracy only evaluate the sign of the reward difference, but not the scale, which could be important during policy optimization. Additionally, the test set may also be contaminated and hence the accuracy may not be accurate.
>
> ### Reference
>
> [1] Amini, Afra, Tim Vieira, and Ryan Cotterell. "Direct preference optimization with an offset." arXiv preprint arXiv:2402.10571 (2024).
>
> [2] Gao, Yang, Dana Alon, and Donald Metzler. "Impact of preference noise on the alignment performance of generative language models." arXiv preprint arXiv:2404.09824 (2024).
>
> [3] Cui, Ganqu, Lifan Yuan, Ning Ding, Guanming Yao, Wei Zhu, Yuan Ni, Guotong Xie, Zhiyuan Liu, and Maosong Sun. "Ultrafeedback: Boosting language models with high-quality feedback." arXiv preprint arXiv:2310.01377 (2023).

---

> > ### Author Response · Authors · 2024-08-13
> > **Any Further Questions**
> >
> > Dear Reviewer hEuH,
> >
> > Thank you again for the detailed and insightful review. As the discussion period is ending, we are wondering if you have any further questions or comments we can address. If our rebuttal has addressed your concerns, we would sincerely appreciate it if you would consider raising your score.

---

> ### Comment · Area_Chair_QvHz · 2024-08-12
> **Please respond to author's rebuttal**
>
> Please respond to the authors' rebuttal to acknowledge that you have read it, and let them know if they've successfully answered your questions and you are willing to change your score, or if you need additional clarification. The discussion period is ending tomorrow, so to give authors sufficient time to respond please try to respond to their rebuttal today.
>
> Currently, while reviewers unanimously agree on accepting this paper (scores are 5,5,6,6), the scores are also low, which means it may not reach the bar for acceptance unless some reviewers increase their score.

---

### Author Rebuttal · Authors · 2024-08-07

We would like to thank all the reviewers for the valuable feedback! Before we answer to each of the reviewers individually, we address common concerns and present our newly added results below:

>**1. Is it possible to run on modern preference dataset like HelpSteer2, Nectar or UltraFeedback, and report the Arena-Hard score as
the evaluation?**

We agree that using modern preference datasets, such as UltraFeedback, would enhance the credibility and validity of our paper. We have trained the model on UltraFeedback and compared its responses against HH SFT on HH dataset and against GPT-4-0314 on Arena-Hard dataset. Due to the limited time during the rebuttal period, we used HH SFT (with Llama2 7b as the backbone model) for our instruction-tuned model. The results are shown in Table 6 in the attached file.

As shown, UltraFeedback improves the performance of both standard DPO and $R^3M$-DPO, with $R^3M$-DPO maintaining its lead over standard DPO on HH dataset. This finding still holds true on Arena-Hard dataset as well. Here, we note that Llama2 7b-chat model has an Arena-Hard score of 4.6. The lower performance compared to Llama2 7b-chat model would be due to using a weaker instruction-tuned model (HH SFT) and the fact that Llama2 7b-chat was trained on a larger and more diverse set of preference data than UltraFeedback. We will revise the manuscript to include these new experimental results.

>**2. Error bars are expected in Table 1.**

Due to computational restrictions, we used a single run for the experiments in Table 1. We agree with your suggestion and have now performed the same experiments with 3 different seeds. The updated results, including the error, are shown in Table 7 in the attached file.

>**3. More ablations along the extremes would help ground the experimental results further and increase the quality of the work.**

We add additional experimental results with extreme noise parameters for the three different noise models on the HalfCheetah task. For the stochastic and myopic noise models, we control a noise-specific parameter that indirectly affects the perturbation rate, so controlling the perturbation rate itself to the extreme would be challenging for these cases. The results are shown in Tables 2, 3, and 4 of the attached file, corresponding to the three different noise models.

For stochastic and irrational noise settings, we observe that model training is completely disrupted for $\tau =100.0$ and $p = 2/3$. However, in the myopic setting, the performance drop is significantly less severe.

>**4. The paper lacks an experiment for formal RLHF, such as Proximal Policy Optimization (PPO), in the context of text generation. Including such an experiment could provide a more comprehensive evaluation of the proposed method.**

Thanks for pointing out the absence of an RLHF baseline in the natural language generation task. We understand the importance of experimenting with PPO instead of DPO to fully comprehend the impact of our method on RM's signal. Due to resource constraints, we prioritized experiments requiring PPO for robotic control tasks. Conducting PPO experiments for natural language generation tasks presents significant computational challenges, including the need to store and manage both reward and critique models (which are LLMs). PPO also involves more training steps and numerous hyperparameter tunings, making it difficult to implement within a resource-limited environment. Therefore, we opted for DPO as an alternative in our submission.

Due to the limited resources and the need to run other experiments, we are currently unable to provide PPO results for natural language generation tasks. Nonetheless, we will try our best to update the results before the discussion deadline.

---

### Decision · Program_Chairs · 2024-09-25

**Decision:**

Accept (poster)

**Comment:**

Reviewers unanimously agree the paper should be accepted (5,5,6,6), with an average rating of 5.5. The paper looks at discarding incorrect human feedback data as outliers, which may have important practical implications for SOTA LLM training.

Strengths:
- well-written
- well-motivated; “raises an important issue in RLHF” (XAMr)
- “Empirical results are generally comprehensive, across two domains, and a hand-full of baselines are considered.” (hEuH) “both robotic control and natural language generation tasks” (XAMr)
- “provides a straightforward and easy-to-use L1 regularization algorithm” (m2Ct); “The method is straightforward” (ugMr)

Weaknesses:
- Some perceived similarity to "Direct Preference Optimization with an Offset" (ODPO)
- Need error bars in Table 1
- Need to use a “modern preference dataset like HelpSteer2, Nectar or UltraFeedback, and report the Arena-Hard” (m2Ct); “try a cleaner dataset” (XAMr)
- Need to include PPO fine-tuning on language models

The author rebuttal provided new results on the UltraFeedback RLHF dataset, and more random seeds, as requested. It also adequately rebutted the similarity to ODPO. I agree with the authors’ explanation about why it’s difficult to provide PPO fine-tuning results on language model (many related papers on RLHF do not provide such fine-tuning results).

In response to the rebuttal, one reviewer (m2Ct) raised their score to a 6, and the remaining 3 did not change their score. One reviewer (hEuH) who gave a score of 5 did not acknowledge the authors’ rebuttal, in spite of it directly addressing their concerns, contributing to a more borderline score for this paper.

Given all of this information, I recommend accepting the paper.